# Binding of Ca^2+^ Ions to Alkylbenzene Sulfonates: Micelle Formation, Second Critical Concentration and Precipitation

**DOI:** 10.3390/ma16020494

**Published:** 2023-01-04

**Authors:** Adél Anna Ádám, Szilveszter Ziegenheim, László Janovák, Márton Szabados, Csaba Bús, Ákos Kukovecz, Zoltán Kónya, Imre Dékány, Pál Sipos, Bence Kutus

**Affiliations:** 1Department of Organic Chemistry, University of Szeged, H-6720 Szeged, Hungary; 2Department of Physical Chemistry and Materials Science, University of Szeged, H-6720 Szeged, Hungary; 3Department of Applied and Environmental Chemistry, University of Szeged, H-6720 Szeged, Hungary; 4Department of Inorganic and Analytical Chemistry, University of Szeged, H-6720 Szeged, Hungary

**Keywords:** linear alkylbenzene sulfonate, calcium, surface tension, CMC, critical concentration, precipitation

## Abstract

Anionic surfactants, such as sodium linear alkylbenzene sulfonates (NaLAS), are utilized in various fields, including industry, household, and agriculture. The efficiency of their use in aqueous environments is significantly affected by the presence of cations, Ca^2+^ and Mg^2+^ in particular, as they can decrease the concentration of the surfactant due to precipitation. To understand cation–sulfonate interactions better, we study both NaLAS colloidal solutions in the presence of CaCl_2_ and precipitates forming at higher salt concentrations. Upon addition of CaCl_2_, we find the surface tension and critical micelle concentration of NaLAS to decrease significantly, in line with earlier findings for alkylbenzylsulfonates in the presence of divalent cations. Strikingly, an increase in the surface tension is discernible above 0.6 g L^–1^ NaLAS, accompanied by the decrease of apparent micelle sizes, which in turn gives rise to transparent systems. Thus, there appears to be a second critical concentration indicating another micellar equilibrium. Furthermore, the maximum salt tolerance of the surfactant is 0.1 g L^–1^ Ca^2+^, above which rapid precipitation occurs yielding sparingly soluble CaLAS_2_∙2H_2_O.

## 1. Introduction

One of the most commonly applied anionic surfactants are long-chain alkyl aryl sulfonates [1], as they can be synthesized from cheap raw materials [1,2] and their design can be tailored to the purpose of application. Among sulfonates, commercially available sodium dodecyl benzenesulfonate (NaDBS) is by far the most studied. It is environmentally advantageous for its faster biodegradation as compared to branched isomers owing to its long aliphatic chain [3], which also results in a very low critical micelle concentration (CMC), i.e., 1.2–2.9 mM [4,5,6,7,8,9,10,11,12,13,14,15]. Moreover, its ability to effectively decrease the interfacial tension (IFT) between an apolar and polar phase, e.g., oil and water, has been reported in earlier works [13,15,16,17,18].

The behavior of surfactants in an aqueous environment depends on a number of physicochemical parameters, such as temperature and salt concentration. Sulfonates can be tolerant to very high temperatures (up to 200 °C) as compared to sulfate detergents [1,19]. However, their usage is restricted to rather low-salinity systems due to their propensity for precipitation [1,19,20,21,22,23]. Beside soap formation, salts have a large impact also on other critical surfactant properties, such as adsorption [24,25,26], transport [27], as well as bubble and foam stability [28,29]. Thus, understanding salt–surfactant interactions is indispensable when assessing the efficiency of surfactants in altering certain physicochemical properties.

In particular, a marked decrease in the surface tension as well as CMC with increasing salt concentration is observed at the presence of electrolytes at a fixed temperature [5,9,12,13,14,15,30,31,32,33,34,35]. For NaDBS as well as other sulfonates, this stems from (1) the reduction of repulsive forces acting between negatively charged head-groups as a result of cation–sulfonate interactions, and from (2) the decreased hydration of surfactant molecules owing to strong cation hydration, which in turn promotes hydrophobic interactions [14,15,31,34,35,36]. The interplay of these two effects leads to the well-known order of ‘salting-out’, observed first for proteins by Hofmeister [15,37].

As for DBS^–^, ions with higher charge density tend to salt out the surfactant at a lower salt concentration. That is, they more readily induce micellization and, eventually, precipitation [13,24,28,29,30]. Indeed, the order of NH_4_^+^ < Na^+^ < Mg^2+^ < Ca^2+^ has been found for precipitating NaDBS from an aqueous solution [15], which is a reversal of the cationic Hofmeister series. The solubility of Ca^2+^ alkyl aryl sulfonates below CMC has been quantified in terms of thermodynamic products [38,39,40]. Based on the thus-obtained equilibrium solubility of Ca^2+^, ca. 1.3–1.7∙10^–4^ M [38,39,40], the calcium salt has a markedly lower solubility as compared to the Na^+^ one, 1.4–4.6∙10^–3^ M [39,40].

The above examples illustrate the high propensity of Ca^2+^ for surfactant precipitation, which may in turn lead to a significant loss of sulfonate surfactant. Thus, characterizing Ca^2+^–surfactant interactions is important to describe the overall performance of a given surfactant in industrial systems. To this end, we study the effect of CaCl_2_ on the properties of sodium linear alkylbenzene sulfonate (NaLAS) aqueous solutions with varying cation and surfactant concentrations in a wide range, both below and above CMC. We find upon increasing the concentration of Ca^2+^ ions a marked decrease in the air–water surface tension as well as CMC in dilute solutions (0.01–0.1 g L^–1^ Ca^2+^). Interestingly, there appears to be a second critical concentration at high surfactant concentrations, associated with another chemical equilibrium. Further enhancement of the amount of CaCl_2_ yields the sparingly soluble CaLAS_2_∙2H_2_O salt, suggesting the maximum salt tolerance of NaLAS to be ~0.1 g L^–1^ Ca^2+^.

## 2. Materials and Methods

### 2.1. Materials

Calcium chloride dihydrate (CaCl_2_·2H_2_O, ACS reagent grade; VWR, Radnor, PA, USA), magnesium chloride hexahydrate (MgCl_2_∙6H_2_O, ACS reagent grade; Merck, Darmstadt, Germany), sodium ‘dodecylbenzensulfonate’ (NaDBS, technical grade; Sigma-Aldrich, St Louis, MO, USA), ethylenediamine tetraacetic acid disodium salt dihydrate (Na_2_EDTA∙2H_2_O, analytical grade; Reanal, Budapest, Hungary), and methanol (HPLC gradient grade; VWR, Radnor, PA, USA) were used as received. The metal-ion content of the hydrated salts was checked by complexometric titration using Na_2_EDTA as titrant.

As for NaDBS, it is known to be composed of sulfonates with different chain lengths [34]. Thus, its exact composition was determined via high-performance liquid chromatography coupled with mass spectrometry (HPLC-MS; for technical details, see below). Based on the relative peak counts, the highest fraction is undecylbenzene sulfonate (43%) followed by dodecylbenzene sulfonate (27%) with the average molar mass being 336.81 g mol^–1^ for the sodium salt; see the mass spectrum in Figure 1. Therefore, this compound will be referred to as sodium linear alkylbenzene sulfonate, NaLAS, throughout this work.

### 2.2. Preparation of Aqueous Samples and Precipitates

All samples were made by using deionized water, which was produced by reverse osmosis and was further purified by UV irradiation, using a Puranity TU3 UV/UF+ system (VWR). Furthermore, samples were always freshly prepared before tensiometric and dynamic light scattering (DLS) measurements, since long-term aggregation of micelles or particle cohesion may alter the samples properties significantly.

First, solution with an appropriate concentration of Ca^2+^-ion (0.01–5.00 g L^–1^) was prepared from CaCl_2∙_2H_2_O. This solution was then filled into a volumetric flask containing the necessary amount of solid NaLAS. The thus-obtained aqueous mixtures (*c*_NaLAS_ = 0.05–5.00 g L^–1^) were stirred for 24 h at room temperature. To avoid the incidental adsorption of either Na^+^, Ca^2+^, Cl^–^, or LAS^–^ ions onto the glass surface of the flask, plastic vessels were used to store all colloidal solutions and filtrates. In case of precipitate formation, the samples were filtered, washed with 20 mL water, and dried. In addition, several precipitates were calcined at 900 °C for 24 h or at 1000 °C for 16 h under air in a Nabertherm L9 furnace (Lilienthal, Germany).

### 2.3. Experimental Methods

The composition of the surfactant was deduced using a 1260 Infinity II HPLC setup coupled to a G6125B LC-MSD (mass selective detection) from Agilent (Santa Clara, CF, USA), applying electrospray ionization (ESI). For analysis, an ~0.05 g L^–1^ (50 ppm) NaLAS aqueous solution was prepared and a water: methanol mixture at 85:15 volume ratio was used as eluent. The mass spectrum was carried out in negative-ion mode, scanning the *m*/*z* region of 150–445.

The surface tensions at the air–water interface and the critical micelle concentration (CMC) values of a set of surfactant solutions with different electrolyte concentration were determined at (25.0 ± 0.1) °C using a tensiometer (type K100; Krüss, Hamburg, Germany), with the aid of the Wilhelmy plate method. In the tensiometer, a platinum–iridium ring is suspended from a torsion balance, and the force (in mN m^–1^) required to pull the ring off the surface is measured. Here, additional experiments were performed for solutions prepared from MgCl_2_∙6H_2_O as described above.

The average micelle size in a set of samples was determined with a Malvern NanoZS dynamic light scattering (DLS) instrument (Malvern, UK) operating with a 4 mW helium–neon laser light source (*λ* = 633 nm). The measurements were performed in back-scattering mode at 173° and at (25.0 ± 0.1) °C. The samples were stirred for 24 h prior to the measurements. For each sample, three to four repetitions were carried out and the size was taken as the arithmetic mean of the volume-averaged hydrodynamic diameters.

For several samples with varying *c*_NaLAS_ (0.05–5.00 g L^–1^) at *c*_Ca2+_ = 0.01 g L^–1^, the apparent absorbance arising from light absorption and light scattering due to colloidal particles was measured in the UV range (*λ* = 200–320 nm, 1 nm optical resolution) at (25.0 ± 0.1) °C, using a Specord 210 PLUS spectrophotometer (Analytik Jena, Jena, Germany).

The X-ray diffractograms of the precipitated as well as calcined solids were obtained using a MiniFlex II type diffractometer from Rigaku (Tokyo, Japan) in the 2*θ* = 4–60° range with 4°/min scanning speed, using CuKα radiation source (*λ* = 1.5418 Å).

FT-IR measurements were carried out using a JASCO FT/IR-4700 spectrometer (Tokyo, Japan). Spectra were acquired between 4000 and 500 cm^–1^ by the accumulation of 256 scans at a resolution of 4 cm^–1^. The spectrometer was equipped with a ZnGe attenuated total reflectance accessory and deuterated triglycine sulfate detector.

The Ca^2+^:Na^+^ molar ratios in the solids were determined by energy-dispersive X-ray (EDX) spectroscopy. To record the spectra, a Röntec QX2 spectrometer (Berlin, Germany) equipped with Be window and coupled to a Hitachi S-4700 scanning electron microscope (Tokyo, Japan) was used at 18 kV acceleration voltages. For each solid, spectra were taken at least four different spots to obtain realistic representation of the elemental distribution.

## 3. Results and Discussion

We started our study with scoping experiments to separate the concentration regions based on the appearance of precipitate. Accordingly, the samples can be divided into two groups: at *c*_Ca2+_ ≤ 0.1 g L^–1^ for all surfactant concentrations (0.01–5.00 g L^–1^), the solutions are transparent or opalescent without the formation of filterable precipitates, whereas at *c*_Ca2+_ > 0.1 g L^–1^ for all surfactant concentrations, a solid appears immediately or after a few hours of stirring (the total equilibration time was 1 day). In possession of this information, we analyzed the colloidal and solid systems according to different aspects to get insights into the behavior of the overall system.

### 3.1. Variation of the Surface Tension and CMC in the Colloidal Regime

First, we studied the effect of Ca^2+^-ion concentration on the CMC of the aqueous samples via surface tension measurements, shown in Figure 2a. As for the pure surfactant, a general trend shows a continuous decline in surface tension with increasing NaLAS concentration until the CMC is reached, where the interface becomes saturated, and surface tension remains almost constant. Upon addition of CaCl_2_, we find the surface tension to decrease markedly. This observation is in line with earlier reports [13,33] and is often elucidated in terms of formation of salt bridges: binding of Ca^2+^ ions screens the electrostatic repulsion between spatially adjacent head-groups, thus facilitating the migration of surfactants to the air–water interface [15,34,35,36,41,42].

Regarding CMC, we obtained 668 mg L^–1^, i.e., 2.0 mmol L^–1^ for neat NaLAS, which falls in the range published in the literature, 1.2–2.9 mM [4,5,6,7,8,9,10,11,12,13,14,15]. It is worth mentioning that when the C12 chain is the dominant component, the CMC is around 1.5 mM [13]. The higher value found here is consistent with the C11 component being the main fraction, since decreasing alkyl chain length gives rise to higher CMC for a series of homologues [4,30]. When CaCl_2_ is present, we observe a marked reduction in the associated CMC, similar to the surface tension: the value found for neat NaLAS, 668 mg L^–1^ (2.0 mmol L^–1^), decreases to 7 mg L^–1^ (0.02 mmol L^–1^) in the presence of 0.1 g L^–1^ Ca^2+^, depicted in Figure 1b. In fact, the decrease has a similar magnitude as found earlier [13]. Such variations have already been reported in the literature [9,13,15,33] and are attributed to the ‘salting-out’ effect: the decrease in electrostatic repulsions between surfactant molecules due to cation binding promotes their aggregation, thus lowering the CMC [31,34]. Furthermore, salts of higher valency render the physicochemical environment less hydrophilic for surfactant molecules (due to the competition for water molecules), thereby having a pronounced impact on the surface activity of LAS^–^ ions and causing micellization at lower surfactant concentrations [35]. It is also seen in Figure 2b that the decrease in CMC is exponential, becoming minor above *c*_Ca2+_ = 0.05 g L^–1^. Such variation of CMC can be expected on the basis of theoretical considerations [30].

Surprisingly, an increase is observed in the surface tension at higher NaLAS concentrations (>0.6 g L^–1^), only in the presence of Ca^2+^ ions (Figure 2a). Such an increase might be indicative of a second CMC, corresponding to a transition from spherical to ellipsoidal, cylindrical or rod-like aggregates, supported by both experiments and simulations for neat surfactants [8,10,43,44]. In fact, a second CMC of 6.9 mM has been determined for NaDBS [8]. The associated micellar shape transformation has been experimentally observed in the presence of salts for both cationic and anionic surfactants [34,45,46,47,48]. However, the appearance of this putative second CMC, based on the inset of Figure 2a, shifts to higher surfactant concentrations with increasing electrolyte concentration, thereby showing the opposite trend as for (the first) CMC. Consequently, in addition to the binding of Ca^2+^ ions to the micelles [49], another chemical equilibrium might be at play: at high NaLAS concentrations with higher LAS^–^:Ca^2+^ molar ratios, smaller associates or even simple CaLAS^+^ ion-pairs might be formed. The latter is well known in the case of SO_4_^2–^ ion [50,51], which is an oxygen donor-ligand, similar to sulfonate. In such a scenario, the more stable Ca^2+^-bound micelles at higher salt concentrations would require higher surfactant concentration to shift the equilibrium toward particle disaggregation, consistent with our observations. Nevertheless, the present experimental data are not conclusive enough to elucidate this phenomenon, which deserves further investigation.

To support these qualitative findings, we carried out tensiometric measurements also in the presence of Mg^2+^. The concentration of Mg^2+^ in each sample, ranging from 0.006 to 0.06 g L^–1^, was chosen such that it matches the concentration of Ca^2+^ in terms of molarities. The obtained surface tension curves are shown in Appendix A. Accordingly, the Mg^2+^/LAS^–^ systems possess the same features for the surface tension as the Ca^2+^/LAS^–^ ones, including the appearance of the second critical concentration (Appendix A) and the steep decrease in the CMC (Appendix A). However, one notable difference is that even for 0.06 g L^–1^ Mg^2+^, the surface tension remains above 50 mN m^–1^ at the lowest surfactant concentration (~2 mg L^–1^), which is much higher than the value of ~32 mN m^–1^ obtained for 0.1 g L^–1^ Ca^2+^ (Figure 1a). This difference suggests much stronger cation–sulfonate interactions for the latter divalent ion, which in turn has a larger effect on the structure of the surfactant monolayer at the air–water interface. Indeed, the perturbation of the surface follows the order of Ca^2+^ > Mg^2+^ > Na^+^, as suggested by density functional theory calculations [52].

### 3.2. Variation of the Average Particle Size at the Precipitation Boundary

The impact of the binding of Ca^2+^ ions on the average particle sizes in the colloidal region has been probed by dynamic light scattering, depicted as volume-averaged average diameters in Figure 3a. In neat surfactant solutions, we obtained very large values (~200 nm) for *c*_NaLAS_ ≤ 0.3 g L^–1^, which seem to be inconsistent with previous experimental and simulation results suggesting the diameter of NaDBS micelles to be 4–6 nm [34,53,54]. It is possible that the high values are experimental artefacts, probably associated with the very low surfactant concentrations not suitable for DLS detection.

In the concentration range of *c*_NaLAS_ = 0.45–0.67 g L^–1^, the obtained diameters drop significantly, which is in broad agreement with the determined CMC (668 mg L^–1^, Figure 2b). At higher concentrations, we observe the diameter to further decrease, reaching ~5 nm at *c*_NaLAS_ = 1.1 g L^–1^. Such a continuous decline has already been observed for NaDBS, i.e., ~16 nm (*c*_NaDBS_ = 6.3 g L^–1^) to ~10 nm (*c*_NaDBS_ = 69 g L^–1^), in the presence of 0.2 M NaCl [53]. However, this decrease was assigned to the breakdown of the Stokes–Einstein equation used to evaluate the DLS data. Indeed, the micelle size tends to increase with concentration for surfactants as a result of increased aggregation [34,55]. Nevertheless, the diameters obtained in this work agree qualitatively with the expected micellar dimensions [34,53,54]. We also note that the corresponding polydispersity indices are rather high (0.3–0.5; see Appendix A), reflecting the somewhat broad size distribution of the surfactant owing to the different alkyl chains.

In the presence of CaCl_2_, the samples become opaque already at low surfactant concentrations with particle diameters clearly exceeding ~200 nm. That is, dissolved species aggregate to a high degree signaling the onset of precipitation, which in turn can be explained by the very low thermodynamic solubility of alkyl aryl sulfonates [38,39,40]. This observation, however, appears to contradict our surface tension measurements, where no precipitation was observed. This contradiction can be resolved by the different timescales of the two experiments: during surface tension measurements—carried out as titrations—the solution was equilibrated only for a short time at each composition, whereas all samples were stirred for one day prior to light scattering detection, allowing the particles to aggregate. This is in line with a previous observation that precipitation starts only after several days at low metal-ion concentrations [38]. In addition, the lowest surfactant concentrations might be problematic to obtain accurate values with DLS.

Strikingly, the samples become transparent upon further increasing the surfactant concentration, which is consistent with the well-known redissolution of poorly soluble sulfonate salts in more concentrated surfactant solutions [38,39,40]. For instance, we observe transparency at *c*_Ca2+_ = 0.01 g L^–1^ and *c*_NaLAS_ = 0.625 g L^–1^, which was confirmed by the drop in the absorbance as well; see Appendix A. At a tenfold concentration of Ca^2+^, however, the sample turns transparent only around *c*_NaLAS_ = 5 g L^–1^. That is, higher Ca^2+^ concentrations required larger amounts of surfactant for the cloudiness to disappear. This trend is similar to the above regarding the reversal of the surface tension: higher *c*_Ca2+_ required higher *c*_NaLAS_ for *γ* to increase (Figure 2a, inset). The similarity between the two trends suggests that there appears to be a second critical surfactant concentration, corresponding to another chemical equilibrium. As for the tensiometric curves, we estimate this critical concentration as the minimum of *γ* (Figure 2a, inset). In the case of the DLS samples, we take this concentration as the average of those belonging to the most concentrated cloudy sample and the most dilute transparent one. A comparison of these estimates from the two methods indeed shows strong correlation (except for *c*_Ca2+_ = 0.075 g L^–1^); see Figure 3b.

In conclusion, the two phenomena, i.e., the increase in the surface tension and the disappearance of cloudiness, stem probably from the same molecular process, likely to be associated with the collapse of large micelles or aggregates. As such, the particle sizes decrease significantly and become very similar to those obtained in the absence of salt at high surfactant concentration (Figure 3a). (Nevertheless, these values should be taken with a grain of salt as the measurements yielded very high polydispersity indices and poorer fits of the correlograms; see Appendix A.)

### 3.3. Characterization of the Precipitates

Having the aqueous phase analyzed, we now turn to the characterization of solids forming at higher metal-ion and surfactant concentrations. We find that above *c*_Ca2+_ = 0.1 g L^–1^, precipitation occurs readily at all surfactant concentrations. That is, the maximum salt tolerance of NaLAS (up to 5 g L^–1^) is 0.1 g L^–1^ Ca^2+^, i.e., 0.28 g L^–1^ CaCl_2_.

The thus-obtained precipitates are all largely amorphous, similar to the poorly crystalline sodium salt, as shown in the X-ray diffractogram in Appendix A. The infrared spectrum of the neat sodium salt matches well the one reported earlier [56], with the following characteristic vibrations in cm^–1^: 2956 and 2870 (–CH_3_ asymmetric and symmetric stretching), 2924 and 2854 (–CH_2_– asymmetric and symmetric stretching), 1601, 1496 and 1408 (aromatic –C=C– stretching), 1461 (–CH_2_ scissoring), 1378 (–CH_3_ symmetric bending), 1182 and 1042 (–S=O asymmetric and symmetric stretching), 1129 and 1012 (aromatic =CH in-plane bending), 831 (aromatic =CH out-of-plane bending), 688 (–SO_3_ bending); see Figure 4.

Upon binding of Ca^2+^ ions, most peak positions remain unaltered (within 4 cm^–1^), with the exception of the –S=O stretching bands which shift from 1182 to 1992 cm^–1^ and 1042 to 1049 cm^–1^, respectively. That is, Na^+^/Ca^2+^ exchange affects mostly the sulfonate moiety, consistent with this group being the metal-ion coordination site. This is consistent with previous calculations for the DBS^–^ anion in the presence of Na^+^, Mg^2+^, and Ca^2+^ ions [52,57], and with Ca^2+^ having a high affinity toward oxygen-donor ligands. In addition, the variation of the –S=O bands are in line with those reported for the incorporation of the DBS^–^ anion in the interlayer gallery of Mg-Al- [56,58], as well as Zn-Fe layered double hydroxides (LDHs) [59]. Nevertheless, the shift to higher wavenumbers is the opposite to that found for LDHs [56,58,59]. Most likely, this is due to the difference between the binding interactions: the sulfonate anion is bound directly to the Ca^2+^ ion in the precipitate, whereas for LDHs, it is connected to the hydrated metal ions via hydrogen-bonding [56,58,59,60]. Moreover, it is seen in Figure 3 that the distance between the positions of the asymmetric and symmetric vibration does not differ significantly (140 vs. 143 cm^–1^), in line with previous findings [56], suggesting similar coordination modes for Na^+^ and Ca^2+^.

Furthermore, the very broad peak around 3000 cm^–1^ in the spectrum of NaLAS, which corresponds to the –OH stretching region, becomes much more intensive for the calcium salt. In parallel, a peak at 1653 cm^–1^ shows up, which belongs to the scissoring mode of water [61,62]. Consequently, the calcium salt is more strongly hydrated than NaLAS. As for calcium salts precipitating from solutions with very different metal-ion-to-surfactant ratios, the spectra are again very similar indicating similar stoichiometries.

We checked the supposed Ca^2+^:LAS^–^ = 1:2 molar ratio by calcining numerous precipitates both at 900 °C (for 24 h) and at 1000 °C (for 16 h). The X-ray diffractograms in Appendix A show unambiguously that the only crystalline solid phase is CaSO_4_ (JCPDS No. 74-2421), with some amorphous CaSO_4_ or residual organics in a few cases (for instance, the precipitate obtained at 5 g L^–1^ Ca^2+^ and NaLAS and calcined at 900 °C; see Appendix A). Consequently, the addition of Ca^2+^ ions to NaLAS solutions yields CaLAS_2_. This is supported by the EDX elemental analyses, showing that—within experimental uncertainty—the solids are essentially free of Na^+^; hence, Na^+^ ions are fully exchanged with Ca^2+^ in the solids. The obtained Na^+^:Ca^2+^ molar ratios are shown in Table 1. (The atomic fraction obtained directly from the measurements are listed as atomic percentages in Appendix A).

These findings are also corroborated by the satisfactory agreement between the mass of the calcined solids and the ‘theoretical’ one, assuming the exclusive formation of CaSO_4_ (which can be calculated by multiplying the weight of the precipitates by the CaSO_4_:CaLAS_2_ molar mass ratio, 136.14/667.72). Nevertheless, differences of 2–11% still remain. Based on the infrared spectra, this difference can be attributed to the presence of hydrating water in the solid phase. Accounting for this water fraction, we obtain 2.3 ± 1.6 water molecules per surfactant unit and thus an average stoichiometry of CaLAS_2_∙2H_2_O. These calculations together with the weight losses are listed in Table 1.

## 4. Conclusions

In this contribution, we studied the interaction between Ca^2+^ and linear alkylbenzene sulfonate (LAS^–^) ions both in the colloidal and heterogeneous systems. We find the CMC (2.0 mmol L^–1^) of the neat surfactant to agree well with the range reported previously, and it is consistent with the C11 chain being the largest fraction. The addition of CaCl_2_ gives rise to a marked decrease in the surface tension and the CMC simultaneously, as a result of cation–sulfonate interactions. Surprisingly, a further increase in the metal-ion concentration results in a sharp increase in the surface tension above ~0.6 g L^–1^ NaLAS, supported by tensiometric curves obtained in the presence of Mg^2+^ ions as well. This ‘second critical’ concentration corresponding to the reversal of the surface tension increases with increasing salt concentration, which is the opposite to the trend commonly observed for CMC. This hints at another micellar equilibrium, being either the transition of rather spherical micelles to ellipsoidal or rod-like aggregates and/or the collapse of micelles, both associated with Ca^2+^-ion binding by the sulfonate group.

These observations are also strongly correlated with the volume-averaged particle diameters as determined from dynamic light scattering experiments. Here, the diameters drop significantly at a certain surfactant concentration, giving rise to macroscopically transparent solutions. These characteristic concentrations are very similar to those corresponding to the onset of increasing surface tension, indicating that both phenomena have the same molecular origin.

Further addition of Ca^2+^ ions to the samples results in rapid precipitate formation at all surfactant concentrations. Consequently, the salt tolerance, i.e., the maximum salt concentration without the appearance of filterable precipitate, is 0.1 g L^–1^ Ca^2+^. All solids are amorphous and identical in their infrared spectra. Na^+^ ions are essentially absent in these precipitates, indicating the exclusive formation of CaLAS_2_. In addition, the weight losses upon calcination for a wide range of initial sample compositions allow us to conclude that CaLAS_2_ precipitates as a dihydrate salt.

## Figures and Tables

**Figure 1 materials-16-00494-f001:**
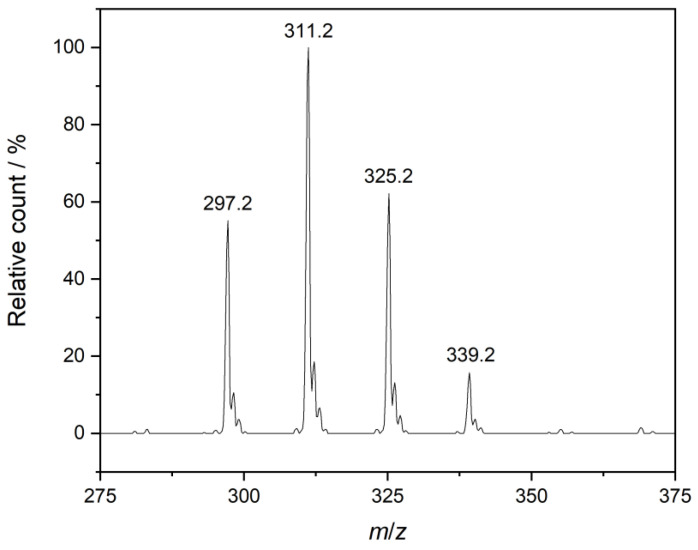
Traces of the negative-ion-mode mass spectrum of an ~0.05 g L^–1^ sodium linear alkylbenzene solution. Peaks indicated follow the order of decyl- (CH_3_(CH_2_)_9_C_6_H_4_SO_3_^–^, *m*/*z* = 297.2), undecyl- (CH_3_(CH_2_)_10_C_6_H_4_SO_3_^–^, *m*/*z* = 311.2), dodecyl- (CH_3_(CH_2_)_11_C_6_H_4_SO_3_^–^, *m*/*z* = 325.2), and tridecylbenzene sulfonate (CH_3_(CH_2_)_12_C_6_H_4_SO_3_^–^, *m*/*z* = 339.2) anions. Based on the relative counts (where the most intense peak of *m*/*z* = 311.2 was taken as 100%), the corresponding molar fractions are 23.7%, 42.9%, 26.7%, and 6.8%, respectively.

**Figure 2 materials-16-00494-f002:**
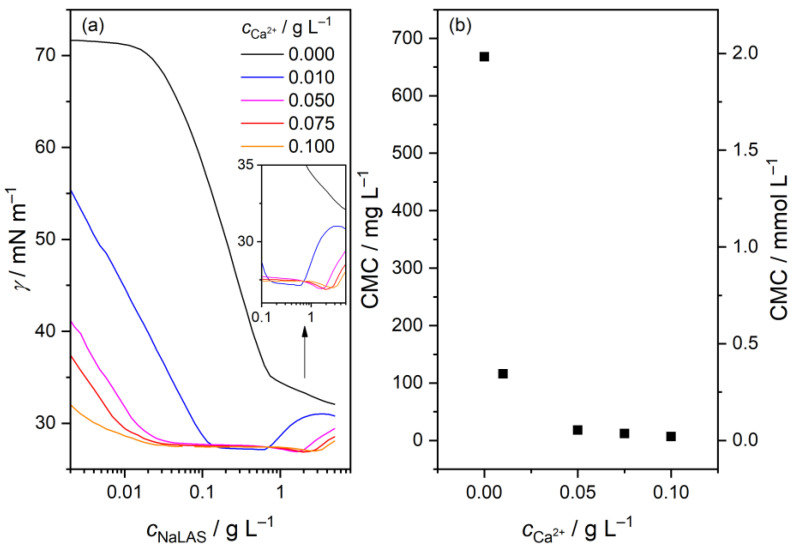
(**a**) Surface tension (*γ*) as a function of sodium linear alkylbenzene sulfonate (NaLAS) concentration, in the presence of CaCl_2_. Inset: zoomed region showing the onset of the increase in *γ*; (**b**) Critical micelle concentration (CMC) in mg L^–1^ (left axis) and in mmol L^–1^ (right axis), as a function of Ca^2+^-ion concentration. Experimental conditions: *c*_Ca2+_ = 0–0.1 g L^–1^, *c*_NaLAS_ = 0.002–5.00 g L^–1^, T = (25.0 ± 0.1) °C.

**Figure 3 materials-16-00494-f003:**
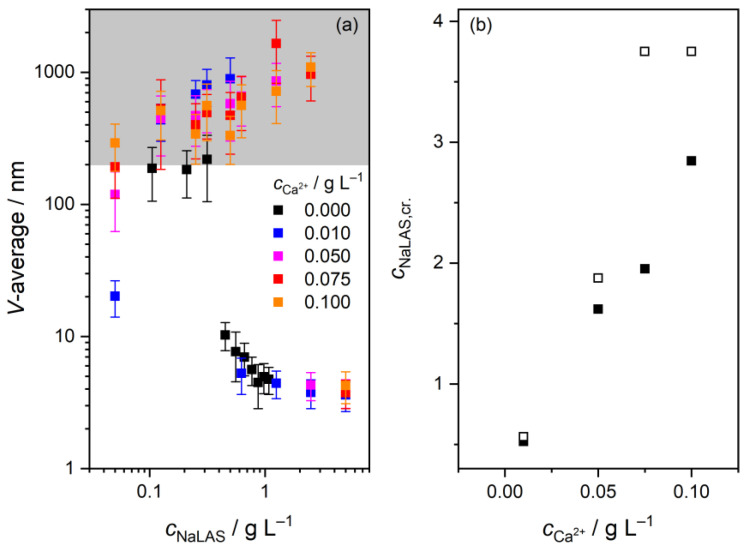
(**a**) Volume-averaged particle diameter as a function of sodium linear alkylbenzene sulfonate (NaLAS) concentration, in the presence of CaCl_2_. The error bars correspond to the highest error of the fit obtained within at least three measurements for each sample. The gray-colored region refers to samples found opaque by visual inspection; (**b**) Apparent second ‘critical’ surfactant concentration (*c*_NaLAS,cr._) as a function of Ca^2+^-ion concentration, as estimated from surface tension (full squares) and dynamic light scattering (empty squares) measurements. Experimental conditions: *c*_Ca2+_ = 0–0.1 g L^–1^, *c*_NaLAS_ = 0.001–5.00 g L^–1^, T = (25 ± 0.1) °C.

**Figure 4 materials-16-00494-f004:**
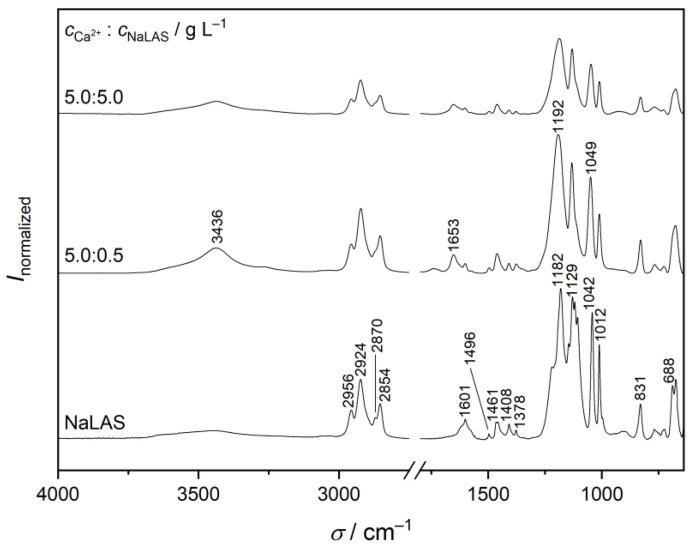
Traces of infrared spectra of sodium linear alkylbenzene sulfonate (NaLAS) and precipitates formed by adding CaCl_2_ to solutions of the surfactant, at two different Ca^2+^:NaLAS weight ratios. Characteristic vibrations of NaLAS taken from Ref. [56] are labelled. Also shown are the stretching (3436 cm^–1^) and scissoring (1653 cm^–1^) vibration modes, respectively, of hydrating water. The intensities were normalized such that the highest value in each spectrum is 1.00.

**Table 1 materials-16-00494-t001:** Concentration of sodium linear alkylbenzene sulfonate (NaLAS) and Ca^2+^ in surfactant—CaCl_2_ dispersions, the Na^+^:Ca^2+^ molar ratio (with standard deviation) in the CaLAS_2_ precipitates forming in the dispersions, weight loss upon calcination (Δ*m*), the difference between the final and theoretical mass assuming the CaLAS_2_ → CaSO_4_ reaction upon calcination (Δ*m*_theo._), the calculated initial H_2_O:CaLAS_2_ molar ratio based on this difference, and the temperature of calcination.

*c*_NaLAS_/g L^–1^	*c*_Ca_^2+^/g L^–1^	*n*_Na_^+^/*n*_Ca_^2+^	Δ*m*/%	Δ*m*_theo._/%	*n*_H_2_O_/*n*_CaLAS2_	*T*_calcination_/°C
0.625	5.00		–81.7	–10.1	4.2	900
2.50	5.00	0.01 ± 0.01	–81.6	–9.59	3.9	900
5.00	0.50	0.1 ± 0.1	–80.2	–3.07	1.2	900
5.00	0.50		–80.1	–2.35	0.9	900
5.00	0.50		–80.0	–2.06	0.8	900
5.00	1.00	0.01 ± 0.01	–80.3	–3.20	1.2	900
5.00	1.00		–80.1	–2.13	0.8	1000
5.00	1.00		–81.7	–10.3	4.2	1000
5.00	2.00	0.05 ± 0.02	–80.2	–3.02	1.2	900
5.00	2.00		–80.0	–1.78	4.9	1000
5.00	2.00		–82.0	–11.7	0.7	1000
5.00	3.00	0.04 ± 0.02	–81.3	–8.22	3.3	900
5.00	4.00	0.02 ± 0.02	–81.3	–8.34	3.4	900
5.00	4.00		–80.6	–4.74	1.8	1000
5.00	5.00		–81.9	–11.6	4.7	900
5.00	5.00		–80.0	–1.7	0.6	1000
5.00	5.00		–80.1	–2.1	0.8	1000

## Data Availability

Not applicable.

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
