# Peer review of "Binding of Ca2+ Ions to Alkylbenzene Sulfonates: Micelle Formation, Second Critical Concentration and Precipitation"

_materials, 2023, doi:10.3390/ma16020494_

Round 1

Reviewer 1 Report

This manuscript is conducted to study Ca2+ influence on the micelle generation, second critical concentration and precipitation of alkylbenzene sulfonates. Some interesting results are obtained in this study but in my opinion, this article only provides superficial analysis for some complicated questions and the related explanations or discussions are not thorough enough. I recommend that more elaborate analysis should be supplemented to make the conclusions more completed. Please give a detailed improvement about your investigation. Only some direct and simple conclusions are obtained from simple analysis methods. Here are some detailed comments:

1.     Line 38, The full name of “IFT” should be provided, when it was introduced at first time.

2.     Line 89, What kind of substance it is for the peak at 339.2 m/z? This seems not to be considered into average molar mass. Besides, what does the relative count of ordinates mean in Figure S1? Please give the explanation, thank you.

3.     Line 190-195, What is the significance of studying the second micelle concentration? What applications can it have? Furthermore, CaLAS+ ion-pairs seems not to be confirmed in the context and the related analysis such as XPS may be necessary to be supplied.

4.     Line 202, If the solubility product can be provided, when sodium salts of sulfonates are sparingly soluble?

5.     Line 205, is there any more suitable measurement methods to replace DLS detection? And if the particle size was not analyzed in detailed, it would be a little superficial and cannot convince readers.

6.     Line 276-277 “That is, Na+/Ca2+ exchange affects mostly the sulfonate moiety, consistent with this group being the metal-ion coordination site.” and 282 “In parallel, a peak at 1653 cm–1 shows up, which belongs to the scissoring mode of water.”, some corresponding literatures should be added to give supports reported by the authors.

7.     Line 292, where is the result of “the EDX elemental analyses”? Both the manuscript and the supplementary materials. Please add it.

Author Response

Our responses are listed in the attachment.

Reviewer 2 Report

Comments to the Author

The work is very interesting but still for better result, I suggest to perform at least one more experiment for another salt containing Mg2+ salt.

1.       Please verify the line “Upon addition of CaCl2, we find the surface tension and critical micelle concentration of NaLAS to decrease significantly, in line with earlier results.’’

2.       Correct the name dodecyl benzenesulfonate (NaDBS)

3.       Please arrange the reference more significantly rather than [4-15].

4.       What is IFT? Please write full form when it first arises.

5.       Explain the term ‘rom soap formation’.

6.       According to me for result, author should perform the same at list one other salt containing Mg2+. Since the efficiency of their use in aqueous environment is significantly affected by the presence of cations, Ca2+ and Mg2+.

I recommend publishing this article after major corrections.

Author Response

(The authors gave the same response as above.)

Reviewer 3 Report

In the publication on calcium ion binding and micelle formation as well as the critical precipitation concentration, the authors discuss the influence of calcium and magnesium ions on the activity of surfactants. Linear alkylbenzene sulfonate ions (LAS–) in both colloidal and heterogeneous systems are tested. The authors conclude that the addition of CaCl2 causes an increase to a marked decrease in surface tension and CMC simultaneously, due to the cation-sulfonate interaction. The high presence of calcium ions results in faster deposit formation at all surfactant concentrations, but this is actually a rather regular base. The researchers also find that CaLAS2 precipitates as the dihydrate salt.

In the opinion of the reviewer, the work is well prepared both in terms of content and in terms of editing. I have no objections and I am applying for acceptance of the work for publication.

Author Response

(The authors gave the same response as above.)

Round 2

Reviewer 1 Report

Thank you for the authors. All the confusions have been explained and I recommend this study to be published as present form. 

Author Response

We thank the Reviewer for their positive evaulation of our revised manuscript.

Reviewer 2 Report

Now the manuscript is modified. It should be published after a minor correction. 

Author should add the following reference given below. 

Volume 69416 February 2018, Pages 7-13; https://doi.org/10.1016/j.cplett.2018.01.029

Author Response

We thank the Reviewer for bringing our attention to the work of S. Roy et al. published in the journal of Chemical Physics Letters. We have read the paper on ion-surfactant association with great interest. This work, however, reports on the binding interaction between an anion and a cationic surfactant in the absence of divalent salts, studied by spectrophotometry and cyclic voltammetry. As such, there is little to no overlap between the scope and experimentation of this paper and our work; therefore, we wish not to add this reference.